# Financial Sustainability and Earnings Management in the Spanish Sports Federations: A Multi-Theoretical Approach

**Juan Carlos Guevara \***, **Emilio Martín** and **María José Arcas**

Faculty of Economics and Business, University of Zaragoza, 50005 Zaragoza, Spain; emartin@unizar.es (E.M.); arcas@unizar.es (M.J.A.)
\* Correspondence: jguevara@unizar.es; Tel.: +34-620-574-171

**Abstract:** The objective of this study is to analyze, from a multi-theoretical framework, whether the managers of the Spanish National Sports Federations (NSFs) apply earnings management using accounting accruals as a measure of managerial discretion; secondly, whether these practices are associated with both the level of dependence on external resources, and the economic and financial control mechanisms exercised by the Superior Sports Council (Consejo Superior de Deportes, CSD) for the granting of public subsidies. The study provides evidence that long-term debt levels and the size of sports federations are determinants of earnings management, with a more accentuated relationship in the case of Olympic and Paralympic sports federations.

**Keywords:** earnings management; accountability; sports federations; nonprofit entities; financial sustainability





## 1. Introduction

In Spain, bankruptcy proceedings, debts to public institutions and a lack of transparency have projected a bad image of the sports sector, a loss of reputation and an understandable suspicion on the part of public opinion about professionalism in the management of this type of entity [1]. This concern is particularly intense for those sports organizations that receive significant amounts of public subsidies to fund their activity.

In the current context of economic crisis and financial austerity, it is particularly timely to question, firstly, whether it is worthwhile dedicating such a large amount of public resources to this type of activity and, secondly, whether these resources are managed correctly. In this sense, the annual accounts are the traditional primary vehicle of accountability to evaluate the transparency and financial performance of organizations. The implementation of the Transparency Law 19/2013 [2] is presented as a regulatory action of the State regarding the management of resources destined for sport that seeks to deal with this distrust by facilitating public access to institutional and economic information from Non-Profit Sports Organizations (NPSOs).

The Spanish sports system is articulated within a public structure whose governing body is the Supreme Council for Sports (Consejo Superior de Deportes, CSD) and a private structure whose governing body are the Spanish Olympic Committee (Comité Olímpico Español, COE) and Spanish Paralympic Committee (Comité Paralímpico Español, CPE) [3]. Sports federations are private nonprofit entities and act as collaborative agents with public administrations [4].

Spain has developed a mixed model in which sports federations receive funds in the form of unconditional grants from the CSD, and revenues from private sources (licenses, sponsors and international organizations). Torres et al. [5], and Martin et al. show a great disparity between NSFs in terms of funding. Thus, public funding provides more than 90% of resources in some federations such as weightlifting and boxing, while in others it does not reach 20% (basketball and Tennis). Public resources have traditionally constituted the main source of funding for non-professional sport-specific grants. The Olympic Sports

Association (Asociación de Deportes Olímpicos, ADO) and Paralympic Sports Association (Asociación de Deportes Paralímpicos, ADOP) Programs or Plans were promoted with the purpose of assigning additional funds to high-level athletes in federations that have more difficulty in attracting private financing.

Additionally, in Spain high-level sports are considered a matter of State interest [4]. Due to this, the development of high-level and high-performance sports resides with the NSFs [6], for which they receive public resources on which they depend to a great extent, and which they must manage efficiently to obtain competitive results [5]. In this sense, sustainability requires a triple bottom line approach, where improvements are pursued in the social, environmental and economic dimensions of performance [7], and NSFs are organizations with social and environmental goals and a growing need to manage their resources efficiently.

A recent CSD report highlights the delicate financial situation of the Spanish sport federations [8], noting that for 2012 only four of the 66 NSFs had a positive working capital or economic result, and only six had positive equity, which places the others in a situation of technical bankruptcy. This situation led the CSD to implement measures from 2013 for those NSFs that have been facing a delicate economic situation, forcing them to draw up a Viability Plan. More than half of the NSFs were subject to these plans, which include strict parameters for cleaning up their financial situation, the non-compliance with which would lead to a penalty in the amount of subsidy granted by the CSD.

These pressures may promote the adoption by some NSF managers of accounting practices aimed at showing a financial representation of these entities in line with the objectives imposed by the CSD. Earnings management (EM) occurs when managers use their judgment in financial reporting and in structuring transactions to alter financial reports to either mislead some stakeholders about the underlying economic performance of the company, or to influence contractual outcomes that depend on reported accounting figures [9].

Since a single theory cannot explain the several responsibilities within a sports organization [10], the theories outlined in this study offer a multi-theoretical approach to analyze to what extent Spanish sports federations apply EM practices, using accounting accruals as a measure of managerial discretion, and secondly, if this is associated both with the level of dependence on external resources and with the economic and financial control mechanisms exercised by the CSD for the awarding of public subsidies.

Furthermore, the study will try to identify if there is a different pattern of behaviour between the Olympic Federations (OFs) and those that are not (Non-Olympic Federations, NOFs). This will lead us to assess accountability in Spanish federated sports and confirm the possibility that managers modify their financial statements when they are monitored on the basis of accounting data [9].

The remainder of this paper is structured as follows: after the introduction, a second section includes the related literature and a statement of the study's hypotheses. Section three develops the empirical analysis; the fourth section presents the results, which are discussed in section five. Finally, the conclusions are stated in section six.

## 2. Previous Literature

In the sports field, the literature on earnings management is scarce and mostly focused on the football industry. Some authors point out the importance of contrasting the results obtained in professional sport with other sectors that share the same idiosyncrasies [11] since generalization about these findings remains an open empirical question [12]. The main contribution of this work is to focus on federated sports, since in non-profit sports organizations one would expect EM practices of different from those observed in professional sports entities, because input and output (or means and ends) are reversed in non-profit and for-profit entities (i.e., non-profits use financial resources as inputs whereas this represents the core output for for-profit) [13,14].

NPSOs give priority to the fulfilment of their sporting objectives over obtaining economic benefits, and in this way satisfy the interests of the governmental organizations that finance them, and for whom success in high-level sport is a determining criterion when allocating resources [15]. This management model focused on sporting performance has made the scarcity of resources a feature associated with the "non-profit" nature of this type of entity [16,17]. In particular, specialized literature has reported the existence of persistent deficits in NSFs [15] as well as in the sports clubs that constitute them [18].

Faced with this situation, the Sport Governing Bodies of different countries have promoted control mechanisms based on an adequate accountability to prevent inefficiencies in the management of public funds by the NPSOs. Some examples are the SFAF, Sport Canada's Sport Funding and Accountability Framework [19], the British Olympic Association (BOA) Financial Times Stock Exchange partnership scheme [20], the programs of the General Secretariat for Sports (GSS) in Greece [21] and the CSD Viability Plans in Spain [22].

### 2.1. Multi-Theoretical Framework

Although there is a variety of theories about governance, previous research suggests that it is a mistake to assume that a single theory applies to any entity, since each theory is focused on specific functions within the organization [10]. This study states that, to analyze governance within an organization, at least three theories must be applied: agency theory, resource dependency theory, and institutional theory [23].

At a general level, the main theory around which the study revolves is classical agency theory. The study raises a classic agency problem in which the principal (Government bodies) establishes measures to monitor the agent's economic performance, but the agent (NSF managers) may try to introduce earning management practices to appear to meet the principal's requirements. However, once the agency problem has been identified, in order to deepen the relationships between Government bodies and NSF managers and, in particular, explain the different behaviors of NSF managers, it is convenient to refer to other complementary theories.

When contextualizing sports federations in this multi-theoretical framework, we can observe that although the "non-profit" nature of this type of entity makes them operate under different paradigms, or even contrary to the "utility paradigm" which is oriented to the search for economic benefit, more and more importance is attributed to the assessment of their financial sustainability as a criterion for the allocation of resources by public partners [20,24]. In this context, one of the main concerns for NSFs is the political costs derived from the economic regulation of the sector.

The State, through legal provisions, has the power to carry out redistributions of wealth among the economic agents of society through the establishment of rates, subsidies, taxes, etc. [25]. Therefore, the set of existing norms at a given moment is the resulting balance between two opposing forces. According to Regulation Theory [26], the most visible organizations tend to be more regulated and accounting information plays an important role in this regulatory process as a monitoring tool. The greater the tax and regulatory aspects, the greater the management incentives to manipulate the result downward, while the entities at risk of not fulfilling their debt contracts will have the incentive to increase the benefit and avoid the negative consequences of not complying with the terms of the agreement [27].

Explaining the regulatory process through the political model represents the essential point of inductive neo-positivism, with which economic theory has made it possible to develop a positive accounting theory, assuming that the interest of management is not necessarily congruent with that of the shareholders. This allows us to understand why the specialized literature highlights, via the Agency Theory [25], the importance of the separation between the managers of sports federations and those who grant them resources from government entities, as an appropriate theoretical framework to understand the management of NPSOs [23]. It would be expected that NPSO managers will choose

accounting policies and measurement bases more aligned to their interests rather than to a true and fair view of the entity [28].

The dependence on public subsidies of federated sport [5,29], and the financial deficits of the sector [18] have contributed to the consolidation over time of a management model characterized by coercive pressure from governments through the financing of NPSOs, significant in countries such as Canada [30], the United Kingdom [31], Australia and New Zealand [32], Spain [29], Germany [33] and more recently Turkey [34] and Greece [15], but less significant in Norway and Belgium [35,36]. In this sense, the influence of the environment will depend on each specific national sports system. For all these studies, the Resource Dependency Theory (RDT) offers an appropriate theoretical platform with which to understand organizations' behavior operating in environments with shifting sources of funds. According to this, an entity's survival depends primarily on its ability to secure and maintain resources and manage associated dependencies [37].

Additionally, success in high-level sports as a determining criterion when allocating resources highlights the importance attributed by governments to competitive results [15]. This could explain why some studies show a higher transfer of resources to the Olympic federations compared to other federations [38]. This fact has generated a culture according to which the financial dimension does not represent a priority and is subordinate to the sporting interests of the organization [39].

Therefore, the importance that current sports development systems attribute to elite sport over mass sport [15], and the responsibility of the NSFs on the development of elite sport [5], justifies the growing literature on the performance of elite sport with public funding and government participation (e.g., [40,41]), as well as on the negative repercussions of a government de-investment in sport [15]. This is because, for NPSOs that depend heavily on state funding, resistance to adopting new regulation, legislation and other forms of change can lead to reduced investment in sport and a loss of valuable state subsidies [23]. All these particularities have led the specialized literature to define NSFs' performance in terms of RDT as: "the ability to acquire and process human, financial and physical resources to achieve the goals of the organization" [13].

Given this reality, it is logical that specific patterns of organizational behavior are replicated in federated sports in different countries and in the NSFs of the same country. Institutional theory defines this type of organizational behavior as institutional isomorphism by arguing that organizations operating within an environment with similar requirements imposed by and expectations related to funding are expected to adopt analogous managerial actions and governance frameworks [42].

This adoption of homogeneous behaviors under the process called isomorphism promotes the stability and survival of organizations, providing them with greater power and institutional legitimacy, by adopting practices oriented to complying with the rules designed by external forces, i.e., coercive isomorphism, or to imitate other organizations in the sector, i.e., mimetic isomorphism [42]. Table 1 provides a taxonomy of some the most representative prior studies.

Thus, this paper provides one of the few qualitative studies that exist in the literature on the management of non-profit sports entities, based on the analysis of the accounting information disclosed by the federations. The rationale is to identify whether, despite not pursuing an economic result, the managers of NSFs are concerned about the information disclosed in their financial statements. Furthermore, despite being an empirical study, it is framed from a multi-theoretical perspective, while most of the other studies are based on a single theory.



**Table 1.** Taxonomy of prior studies.

| Author(s) | Sample | Framework | Design |
|---|---|---|---|
| Edwards et al. [30] | 78 Alberta provincial sport organisations (PSOs) in Canada | IT | Qualitative |
| Nichols et al. [31] | UK sport organisations | VT | Qualitative |
| Shilbury, D.; and Ferkins, L. [32] | 8 Australia and New Zealand sport organisations | IOR | Qualitative |
| Erturan-Ogut, E.; Sahin, M [34] | 8 members of sports federations and members from the Turkish Ministry of Youth and Sport | | Qualitative |
| Skille, E. [35] | 8 sport clubs in the Norwegian Sports City Program (SCP) | IT | Qualitative |
| Vos el al. [36] | Sports clubs belonging to 57 Flanders municipalities | IT, RDT | Quantitative |
| Giannoulakis el al. [15] | 10 Greece sport federations and a representative from the Secretariat of Sport | RDT | Qualitative |
| This study | 58 Spanish sport federations | RDT, IT, AT | Quantitative |

RDT: Resource Dependency Theory, IT: Institutional Theory, AT: Agency Theory, IOR: Interorganizational Relations Theory, VT: Volunteering theory.

*2.2. Formulation of Hypotheses*

Next, we propose the five hypotheses to be tested. The first refers to the possibility that the dependence on public resources in the financing of NSFs is a factor that encourages the introduction of accounting practices aimed at artificially increasing the result.

**Hypothesis 1 (H1)** . *The level of dependence on public funds of NSFs is positively associated with accounting practices that artificially increase the result.*

Most of the literature on the subject shows that sports organizations are characterized by having scarce financial resources, even less than other types of non-profit organizations [43], and in many cases a strong dependence on public contributions [5]. Faced with this situation, a decrease in public subsidies may generate severe financial problems in sports organizations such as in the case of Greece, especially after the reduction of public subsidies after the Olympic Games in Athens in 2004 [15].

The impact of the reduction of public subsidies on non-profit sports organizations has been analyzed in different countries, for example in Canada by Hall et al. [44], in Germany by Langer [45] and in Australia, Great Britain, Finland and Poland through the study by Berrett and Slack [46]. In this regard, in Spain, CSD grants to NSFs experienced a steep year-on-year decline of 24.85% in 2012 and 29.94% in 2013 [8], and consequently this phenomenon is worth analyzing.

This financial dependence on public subsidies gives government sports administrations a certain capacity to implement their sports policies based on their hierarchical position [47]. Willem and Scheerder [48] analyze, in a study that compares federated sport in eleven European countries and two non-European countries, how the government can regulate the NSFs from a combination of coercion, subsidies and collaboration, applied according to different priorities. These, in the majority of countries, are oriented to the development of capacities in the NSFs to generate their own financial resources.

Additionally, Hoye et al., [49] claim that sport has become the object of increasing state regulatory activity, which in the light of Regulation Theory highlights the role of accounting information to understand causal relationships [27]. From this perspective, it is assumed that public authorities monitor the management of sports organizations [50] which could give rise to the manipulation their results upwards in order to convey a better image of financial management and avoid compromising organizational control [37].

The second and third hypotheses analyze whether increased regulation and public control promotes the introduction of creative accounting practices in sports organizations.

A study is required in the case of Spanish NSFs, given that the CSD requires the preparation of a Viability Plan for those entities that present negative net worth or working capital.

**Hypothesis 2 (H2).** *The Equity of NSFs is indirectly associated with the level of EM.*

**Hypothesis 3 (H3).** *The NSFs Working Capital is indirectly associated with the level of EM.*

Both hypotheses are based on the fact that increased state regulatory activity and the economic pressures generated by dependence on external resources [37] promote isomorphisms of the institutional type [42]. In the case of non-profit organizations, this materializes in the adaptation of management techniques from the private sector, with consequent agency problems [25] that, framed by the theory of regulation [27], could lead to the incorporation of EM practices.

In Spain, NSFs with negative net worth or working capital must undergo a Viability Plan, the non-compliance with which would result in a penalty in the amounts allocated in the CSD grants during the years of validity of the same. These measures have resulted in improvements in the financial situation of NSFs [22], but their effect on accounting quality remains to be determined.

The fourth hypothesis will address the possible relationship between the indebtedness of organizations and the application of profit management accounting techniques.

**Hypothesis 4 (H4).** *The level of indebtedness of NSFs is positively associated with the level of EM.*

The level of indebtedness, understood as the sum of short-term liability plus long-term liability [22], is another of the figures observed by the CSD, which although not part of the criteria for adopting a plan viability, is a determining factor of the financial health of NSFs. Given that the behaviour of short-term debt has been analyzed through the Working Capital Fund, it should also be observed whether the management of profit bears any relationship to the long-term debt structure. For this purpose, we have considered the proportions of long-term liabilities to total assets, which allows comparisons between entities of different sizes. In the case of Spanish NSFs, it is observed that, as of 2013, the loans granted by the CSD have been reduced considerably, since they became linked to the implementation of financial improvement plans [8].

The last hypothesis analyzes the relationship between the size of the organization and the application of accounting techniques relating to for-profit management.

**Hypothesis 5 (H5).** *The size of NSFs is indirectly associated with the level of EM.*

According to Orlitzky [51], firm size is positively related to performance and business viability, since it can lead to economies of scale in operations and greater control over external stakeholders and resources, since the potential for regulatory scrutiny increases as companies get bigger and more profitable [52,53].

For these reasons, we hope to be able to confirm in federated sport the same negative relationship between size and the tendency to manipulate the result previously observed in the business sector [52], and in professional sport through the teams of the top professional European football leagues [11,12].

## 3. Materials and Methods

### 3.1. Sample and Data

The last Olympic cycle (2013–2016) was analyzed, the beginning of which coincides with the entry into force of the CSD Viability Plans and the implementation of Transparency Law 19/2013 for NSFs. Thus, the initial sample considered the total number of national NSFs (66), discarding those NSFs that did not provide financial data for the mentioned periods.

The final sample consisted of 52 NSFs for the year 2013, and 58 NSFs for the years 2014, 2015 and 2016, totaling 226 observations. The accounting data has been captured manually from the transparency portals contained in the web pages of the Spanish NSFs. The study uses the information included in the individual annual accounts and their corresponding audit reports. Complementary data have been obtained from the CSD's "other statistics" section for the years 2013 to 2016 [54], and the DEPORTEData database of the Ministry of Education, Culture and Sports [55]. It should be noted that, before 2013, the year in which the Transparency Law 19/2013 was promulgated, no federation had disclosed financial reports on its website.

This fact has represented an advance concerning other studies [15,34,38], since in Spain, thanks to the favorable effect of Transparency Law 19/2013, we have the data for the realization of this study. However, a limitation to its scope has been the lack of previous data that would allow assessment of the behaviour of NSFs before the implementation of Transparency Law 19/2013 and the CSD Viability Plans.

### 3.2. Estimation of Discretional Accruals

Many antecedents in the literature evidence a deterioration in accounting quality, with regulatory interventions that assess the financial position and performance of companies based on the reported accounting information [56]. In the sports field, some studies have used EM as a proxy for accounting quality [11,12].

The most used commonly methodology to detect and measure EM is the estimation of discretional accruals (DA). Accruals are the accounting adjustments between cash flows and income and depend on managerial estimates and assumptions; thus, they are subject to managerial discretion and manipulation.

Discretional accruals (DA) are defined as that part of income or expenses that do not involve cash flows. Indirectly, they are calculated through a balance sheet approach or by the difference between the result and the operating cash flow (CFO). If we assume that the latter are not susceptible to manipulation, the way to alter the result is by accrual adjustments (TACC). In this sense, TACC, through the estimation of standard or non-discretionary accruals (NDA), provide us with a point of reference from which to consider the abnormal or discretionary accrual (DA) as proxies of managerial discretion, reduced to the following base expression:

$$TACC = NDA + DA \qquad (1)$$

where:

TACC = Total accruals.
NDA = Non-discretionary accruals.
DA = Discretionary accruals.

The line of research on earnings management begins with the work of Jones [57], who argues for measuring discretion through the study of TACC, differentiating between the discretionary and non-discretionary parts. To do this, a linear relationship is established between TACC, changes in sales and property, and plant and equipment; hence, the non-discretionary part is that which is given by changes in sales figures and fixed assets.

Discretionary Accruals as a Measure of Earning Management

To measure the level of accounting manipulation, we rely on DA. TACCs are calculated indirectly through a balance sheet approach that assumes that the adjustments for a period are obtained by subtracting the change in current liabilities from the change in current assets that are not cash, for that given period. They can be expressed according to the following approach [58]:

$$TACC_{it} = (\Delta CA_{it} - \Delta Cash_{it}) - (\Delta CL_{it} - \Delta STDEBT_{it}) - DEP_{it} \qquad (2)$$

where:

$\Delta CA_{it}$ = the change in current assets of NSF i in period t;

$\Delta CL_{it}$ = the change in current liabilities of NSF i in period t;

$\Delta Cash_{it}$ = the change in cash and cash equivalents of NSF i during period t;

$\Delta STDEBT_{it}$ = the current maturities of long-term debt and other short-term debt included in current liabilities during period t;

and $DEP_{it}$ = depreciation and amortization expense of NSF i during period t.

All variables are deflated by lagged total assets ($TA_{t-1}$) to control for scale differences.

To differentiate between the discretionary and nondiscretionary part of TACC, we have chosen to use the Jones model [57], according to which the expected adjustments are calculated as a function of the variation in gross income and property, plant and equipment (PPE), which is controlled through depreciation and amortization expense adjustments. All variables in the accruals expectations model are scaled by previous year assets to reduce heteroscedasticity. The approach of the model is as follows:

$$TACC_{it}/TA_{it-1} = \alpha 1\,(1/TA_{it-1}) + \alpha 2(\Delta REV_{it}/TA_{it-1}) + \alpha 3(PPE_{it}/TA_{it-1}) + \varepsilon_{it} \quad (3)$$

where:

$TACC_{it}$ = Total accruals in year t for NSF i.

$\Delta REV_{it}$ = Revenues in year t less revenues in year t − 1 for NSF i.

$PPE_{it}$ = Gross property, plant and equipment in year t for NSF i.

$TA_{it-1}$ = Total assets in year t − 1 for NSF i.

The Jones model [57] has been the basis for multiple subsequent proposals including those models that require ROA (return on active), as in the model of Kothari et al. [59], because profitability does not represent a core objective of non-profit entities. Models that require a cash flow statement, such as the Kasznik [60] model, are also discarded due to the lack of availability of such information, since, in Spain most NSFs, due to their size, do not have an obligation to present a cash flow statement when a balance sheet, statement of changes in equity and an abbreviated report can be formulated [61].

### 3.3. Conditioning factors of Earning Management

In order to test our research hypotheses, we incorporate discretionary adjustments (DA) as a dependent variable in the following model (Generalized Least Squares, GLS):

$$DA_{it} = \alpha_0 + \alpha_1\,RD_{it} + \alpha_2\,E_{it} + \alpha_3\,WC_{it} + \alpha_4\,LTL_{it} + \alpha_5\,SIZE_{it} + \varepsilon_{it} \quad (4)$$

where $DA_{it}$ are the discretionary accruals captured through the error term of the model (3) as a proxy variable for earning management. To test the first hypothesis H1, the $RD_{it}$ variable captures public funds' dependence measured by public resources divided by total revenues. For this, the public funds received by the NSFs are calculated through the sum of the subsidies of the CSD and additional channels such as the ADO and ADOP programs. It is considered that a growing dependence on external resources that compromises the autonomy of the NSFs [37] could be an incentive for manipulation, especially for those NSFs that present financial deficiencies, and for which public subsidies represent a determining means for the fulfilment of objectives [43]. Therefore, a positive coefficient is expected for this variable.

To test hypotheses H2 and H3, the variables $E_{it}$ and $WC_{it}$ are dummies that capture the signs of Equity and Working Capital respectively, taking the form (0) if the value is negative, or (1) if it is positive. It is considered that the NSFs will seek to report positive balances in both cases, since with one of these two figures, or with both simultaneously, negative, the federation must undergo a Viability Plan [22]. We assume that this fact entails an incentive for manipulation, for which a negative coefficient is expected for both variables.

The $LTL_{it}$ variable with which the proportion of Long-Term Debt is represented as a function of Total Assets allows us to contrast our fourth hypothesis H4, according to which profit management will be positively associated with a higher proportion of long-term debt.

To contrast our fifth and final hypothesis H5, Model (4) includes the size ($SIZE_{it}$) of the NSFs as a variable which has been shown to be an essential determinant of earnings management in previous research in the sports field [11,12]. We used the natural logarithm of end-of-year total assets ($LnTA_{it}$), as a measure of the NSF's size, with larger NSFs expected to seek less visibility while diminishing their profits, as the effect of regulatory scrutiny increases as the companies get larger [52,53].

The study also considers that NSFs, as beneficiaries of the ADO and ADOP programs, have the additional need to satisfy the expectations of the taxpayers paying for each program, which could represent more significant incentives to manipulate the accounting figures concerning other NSFs that do not have access to these benefits as a consequence of complying with the Olympic and Paralympic cycles. This fact implies the need to assess the different incentives for the manipulation of the results between Olympic federations (summer and winter) or Paralympic (OF), and those that do not participate in any of these multisport events (NOF). Finally, model (4) is performed separately in order to observe possible differences in behaviour between OF and NOF. For all cases, there was no evidence of multicollinearity in the variables used in the regression model.

## 4. Results

Our first objective is to know if the NSFs manipulate the results of the analyzed exercises based on the information disclosed as a result of the implementation of Transparency Law 19/2013. Hence, we rely on discretionary accrual, whose calculation is based, as explained above, from the estimation of equation (3). An ordinary least squares regression has been applied for each period, and in this way the DA can be obtained through cross-sectional estimates. Table 2 presents the descriptive statistics of the residuals estimated by OLS of equation (3) for each period.

**Table 2.** Descriptive statistics of discretionary accruals (2013–2016).

| Year | n | Mean | Median | SD | Minimum | Maximum |
|------|-----|---------|---------|--------|---------|---------|
| 2013 | 52 | 0.0302 | 0.0166 | 0.2825 | −1.1903 | 0.5773 |
| 2014 | 58 | 0.0170 | −0.0215 | 0.4331 | −1.9321 | 1.0626 |
| 2015 | 58 | −0.0521 | 0.0306 | 0.3151 | −1.0813 | 0.7039 |
| 2016 | 58 | 0.0459 | 0.0741 | 0.3224 | −0.8736 | 0.6031 |

When analyzing discretionary accrual, a predominance of upward adjustments is observed, except for the median for 2014, the year in which Transparency Law 19/2013 came into force. This fact suggests the attempt by NSFs to present positive results in order to avoid the sanctions of the CSD.

### 4.1. Multivariate Analysis

The descriptive statistics of the series (Table 3) show predominantly positive discretionary accruals. Additionally, public funds represent 41.04% of the resources of all NSFs. Regarding the financial parameters of the CSD, 62.39% of the NSFs have a Positive Equity (NE), and 53.54% have a positive Working Capital (WC), which confirms the high percentage of NSFs with NE or with negative WC.

Subsequently, the panel data GLS methodology with random effects is applied according to the result of the Haussman test: Chi-square = 8.90 (*p*-value= 0.1132), which is also consistent with the data structure of the sample (n > t). A panel has been used that incorporates 226 observations from the NSFs that presented data for all the years of the Olympic cycle analyzed. The results are described in Table 4:

**Table 3.** Descriptive statistics of the variables of equation 4 for the research period (2013–2016).

|  | **Mean** | **Median** | **SD** | **Minimum** | **Maximum** |
|---|---|---|---|---|---|
| $DA_{it}$ | 0.0097 | 0.0283 | 0.3443 | −1.9321 | 1.0626 |
| $RD_{it}$ | 0.4104 | 0.4064 | 0.2167 | 0.0000 | 0.9367 |
| $E_{it}$ | 0.6239 | 1.0000 | 0.4855 | 0.0000 | 1.0000 |
| $WC_{it}$ | 0.5354 | 1.0000 | 0.4999 | 0.0000 | 1.0000 |
| $LTL_{it}$ | 0.2743 | 0.0821 | 0.4327 | 0.0000 | 2.3688 |
| $SIZE_{it}$ | 13.2812 | 13.1429 | 1.6760 | 9.0721 | 18.8971 |

$DA_{it}$: Discretional accruals, it's a proxy variable for earning management captured through the error term of the model (3). $RD_{it}$: Resource Dependency measured by public resources divided by total revenues. $E_{it}$: Dummy that capture the sign of Equity. $WC_{it}$: Dummy that capture the sign of Working capital. $LTL_{it}$: Long-Term Debt (proportion of Long-Term Debt as a function of Total Asset). $SIZE_{it}$: is the natural logarithm of end-of-year total assets.

**Table 4.** Results of the estimation of Equation (4).

|  | **Coefficient** | **z** | **Significance** |
|---|---|---|---|
| $RD_{it}$ | −0.0506 | −1.07 | 0.285 |
| $E_{it}$ | 0.0362 | 1.40 | 0.163 |
| $WC_{it}$ | −0.0213 | −0.99 | 0.323 |
| $LTL_{it}$ | 0.0860 | 2.51 | 0.012 ** |
| $SIZE_{it}$ | −0.0122 | −2.53 | 0.011 ** |
| Constant | 0.1682 | 1.98 | 0.048 |

No Cross-sectional dependence based on the Pesaran [62] test, stats = 0.746 (*p*-value = 0.4558). No serial correlation based on the Drukker [63] test, F = 1.695 (*p*-value = 0.1988). The asterisks represent the statistically significant coefficients at the 5% (**) significance level. R-sq = 0.1508.

The proportion of public funds to the total income of each federation ($RD_{it}$) presents a coefficient that is not statistically significant. As a consequence, we reject our first hypothesis (H1), according to which the NSFs with greater dependence on public resources would be more likely to manipulate their results. Regarding the dummy variables $E_{it}$ and $WC_{it}$ with which the signs of Net Equity and Working Capital are captured respectively, they present positive coefficients, contrary to what was expected. However, this is not statistically significant for either of the two cases. For this reason, hypotheses (H2) and (H3) are rejected, according to which the manipulation of the result may be associated with the control mechanisms exercised by the CSD that condition the delivery of subsidies by a Viability Plan when the NSFs show a negative balance in Equity or Working Capital. The $LTL_{it}$ variable with which the proportion of long-term liabilities is represented as a function of total assets presents a positive coefficient and with high statistical significance ($p = 0.012$ **), with which our fourth and last hypothesis is not rejected. (H4). This result allows us to affirm that the NSFs most indebted in the long term manipulate the result upwards.

Regarding the size variable ($SIZE_{it}$) expressed through the natural logarithm of the asset, a negative and significant coefficient is observed ($p = 0.011$ **), which confirms, in line with hypothesis 5, an inverse relationship between the size of the NSFs and the DA, suggesting that larger NSFs engage less in earnings management so as to avoid reporting losses.

### 4.2. Contrast between the Olympic and Paralympic Federations (OF) and the Other Federations (NOF)

When separating the two samples in model (4) to assess the different incentives for the manipulation of the result between OF and those that do not participate in any of these multisport events (NOF), substantial differences between both are evident (see Table 5), observing a greater explanatory power of the model in the OF ($R^2 = 0.1246\%$) than for the NOF ($R^2 = 0.0413\%$).

**Table 5.** Discriminating the results of the estimation of equation (4) between OF and NOF.

| | OLYMPIC & PARALYMPIC FEDERATIONS (OF) | | | OTHERS FEDERATIONS (NOF) | | |
|---|---|---|---|---|---|---|
| | Coefficient | z | Significance | Coefficient | t | Significance |
| $DA_{it}$ | −0.8977840 | −1.24 | 0.214 | −0.0317618 | −0.33 | 0.738 |
| $E_{it}$ | 0.0748924 | 2.00 | 0.045** | −0.0239450 | −0.52 | 0.604 |
| $WC_{it}$ | −0.0077427 | −0.24 | 0.811 | −0.0462061 | −1.33 | 0.183 |
| $LTL_{it}$ | 0.1203841 | 2.41 | 0.016** | 0.0104904 | 0.12 | 0.904 |
| $SIZE_{it}$ | −0.0194904 | −2.24 | 0.025** | −0.0217996 | −1.13 | 0.257 |
| Constant | 0.2555554 | 1.61 | 0.108 | 0.3426228 | 1.37 | 0.171 |
| | R-sq = | | 0.124 | R-sq = | | 0.041 |
| | Number Observations = | | 126 | Number Observations = | | 100 |
| | Number of groups = | | 32 | Number of groups = | | 27 |
| | Fixed-effects | | | Random-effects | | |

No Cross-sectional dependence based on the Pesaran [62] test. No serial correlation based on the Drukker [63] test. The asterisks represent the statistically significant coefficients at the 5% (**) significance level.

In addition to the marked differentiation between OF and NOF, observing a similar behaviour between OF and total NSFs suggests that OF could have influenced this result, since the behaviour of the $LTL_{it}$ variable is replicated with statistical significance ($p = 0.016$ **), according to which the manipulation of the result upwards is a consequence of high levels of long-term debt. The size variable ($SIZE_{it}$) is also statistically significant ($p = 0.025$ **), which confirms that the largest OFs tend to manipulate the result downward as a possible strategy to avoid visibility. Additionally, a direct and statistically significant relationship (**) is observed between the NE variable and the DAs, which had not been observed in the NSF set, and which could suggest a trend for later periods.

### 4.3. Sensitivity Analysis

To resolve the influence of extreme values on the results without a decrease in the sample, all the variables used to estimate the model (3) have been treated by substituting extreme values for more typical values using the Winsorizing technique [64] in order to obtain more robust results.

To examine the robustness of the findings, we performed several sensitivity tests against the model specification. First, we re-estimate DAs using the Jones [57] model modified by Dechow et al. [65], which corrects the variation in revenues for the variation in accounts receivable. This is because the Jones model does not take into account the variations between revenues and receivables, considering that the variation in revenues is non-discretionary, without taking into consideration the possible anticipation of the same (and, with it, the variation in the number of debtors). The results remain unchanged after this modification. Second, we include a constant to provide an additional control for heteroscedasticity and we re-estimate the DAs for both models [57,65], and the results remain qualitatively unchanged. Consequently, our findings are robust to alternative methods for calculating DA.

### 5. Discussion

When analyzing the results obtained under a multi-theoretical framework, we can first verify the presence of isomorphisms [42] as a consequence of the increasing state regulatory activity for the sector indicated in the literature [49]. This behaviour can be seen through the increase in the disclosure of financial information from the enactment of Transparency Law 19/2013, and has allowed overcoming some limitations in scope of previous studies in the sector [15,34,38] in which such information has not been available.

The presence of discretionary accrual confirms in Spanish federated sports that when dominant stakeholders exert pressure in sectors with financial difficulties, it is very likely that the quality of accounting will deteriorate significantly, as previously claimed by Dimitropoulos et al. [11] in professional sport. This fact could be explained by agency problems [25] between NSF managers and government agents. This result could be associated

with the adoption of techniques from the business sector previously observed by Zintz and Vailleau, [66] in the Belgian and French NSFs, as a consequence of a possible trend towards professionalization of managers [67].

The median DA indicates a predominance of upward adjustments (>50%). This behaviour could suggest a trend as a consequence of the entry into force of Transparency Law 19/2013 and the CSD Viability Plans for NSFs. However, the results obtained are not conclusive, since there are no previous data that allow the performance to be compared with periods before 2013, when both regulatory measures were implemented.

In contrast to postulates of the RDT [37], the results of the study do not show any evidence that links earnings management with dependence on public funds (variable $RD_{it}$), nor with the conditions of the CSD for granting them (variables $E_{it}$ and $WC_{it}$), except for the FFOO, in which case a high statistical significance is observed in the variable $PN_{it}$ ($p = 0.045$ **). This fact may be due to the increase in CSD subsidies during the period studied [8]. However, the results of the OF stand out from the rest of the NSFs, a fact that can be understood through previous studies that indicate that the OF receive more resources than other types of federation [38].

In Spain, the OFs are subscribed to the ADO and ADOP programs respectively, whose instrumentalization involves public funds and additional controls that may have an impact on the manipulation of the result. This situation may lead OFs to adopt homogeneous accounting practices (or mimetic isomorphisms) as a consequence of a standard programming based on the Olympic cycle. In this case, the time series analyzed coincides with the Rio 2016 Olympic cycle. This would stimulate the emergence of agency problems [25] between the executives in command of the NSFs and its main creditors, i.e., government authorities. Faced with this situation, as proposed by the RDT [37], the NSFs are called upon to diversify their resource acquisition strategy by seeking additional income such as via commercial activities and private contributions (for example, sponsorship).

The behaviour of the $LTL_{it}$ variable ($p = 0.012$ *), which explains the link between high levels of long-term debt and the use of DAs to increase profits, could be related to the dynamics of some NSFs, especially OFs ($p = 0.016$ *), which commit resources beyond one financial year and can be projected over the four years that each Olympic cycle entails.

Through the size variable ($SIZE_{it}$), the present study provides empirical evidence that larger NSFs seek less visibility, reducing their benefits. This fact confirms that the effect of regulatory scrutiny increases as the companies become more massive [52,53], and in turn corroborates the same behaviour observed in the field of NPSOs' professional sports [11,12]. These findings, added to the marked difference in the behaviour of concerning the rest, are clear evidence of mimetic isomorphisms between NSFs of the same size and type. Additionally, the fact that OFs are more likely to manipulate their results upwards when they are more indebted, or downwards as their size increases, could be related in terms of accountability with the results of previous studies that place OF as less transparent than other NSFs [38].

*Managerial Insights*

This section provides some recommendations based on the key observations for sport policy makers and NSF managers.

The study emphasizes the importance of accountability as an element of control of the performance of managers of non-profit organizations. Traditionally, the performance of sports federations had only been measured by sports results or by their social impact in promoting grassroots sport without paying much attention to the management of public money. However, the gradual implementation of the new public management postulates together with the context of financial crisis and high public deficits have motivated the Spanish Government to place a greater focus on the financial situation of NSFs. The introduction of financing and remuneration mechanisms based on the fulfillment of budgetary and financial objectives creates incentives for managers to introduce earning management practices, deteriorating the quality of accounting information from NSFs. This situation,

which arises as a consequence of the pressure exerted from the CSD, gives rise to the classic agency problem [25]. Multi-year budgeting may help to improve long-term planning by NSFs managers while facilitating the monitoring by governments of fulfillment of viability plans.

Additionally, via the triple bottom line (TBL) approach, this type of policy has contributed to highlighting the importance of the financial pillar as a necessary factor to achieve the traditional social and environmental purposes of NSFs. In this regard, according to the framework of the Theory of Regulation [27], the study proposes the adoption of a management model more focused on the capacity of NSFs to generate their own income to guarantee their financial sustainability and expand the coverage of their social benefits.

## 6. Conclusions

The present study, framed by a multi-theoretical model, is presented as a contribution to the few EM studies on non-profit entities, and more specifically those belonging to the sports sector. The study analyzes the extent to which EM practices are used in Spanish NSFs, with accrual adjustments as a measure of managerial discretion and, secondly, whether this is associated with both the level of dependence on external resources and the economic and financial control mechanisms exercised by the CSD for the granting of public subsidies. Subsequently, whether there are differences between the behaviour patterns of the OF and NOF is observed.

In the first place, a greater disclosure of financial information is observed since the enactment of Law Transparency19/2013. Additionally, the presence of discretionary accrual adjustments confirms a deterioration in accounting quality as a consequence of the pressure exerted by the CSD as one of the main stakeholders of the Spanish NSFs.

Although the study does not provide any significant evidence linking the dependence on external resources, or the financial controls of the CSD, with the manipulation of the results of NSFs, the marked difference between the OF and those that are not could be a consequence of this type of NSF being subscribed to programs (ADO and ADOP) that involve the allocation of public funds. This fact could explain the influence of the pressures exerted by dominant stakeholders, such as the CSD, on the deterioration in the accounting quality of the Spanish NSFs. The study also provides evidence that long-term debt levels and the size of NSFs are determinants of profit management, observing a more robust relationship in the case of OFs.

When interpreting the findings from a multi-theoretical perspective, it is highlighted, first of all in the Agency Theory Framework [25], that the incorporation of regulatory monitoring based on accounting data can lead to differences between managers of NSFs and representatives of government bodies that provide financial support to them, regarding a deterioration in accounting quality. Secondly, within the framework of Institutional Theory [42] all the findings imply coercive isomorphisms before the regulatory framework, and mimetic isomorphisms expressed in the same accounting practices by NSFs with the same level of long-term debt, the same size, or the same type in the case of Olympic and Paralympic (OF). Within the framework of Resource Dependence Theory [37], the influence of the pressures exerted by the dominant stakeholders through financing is confirmed. Additionally, within the framework of the Theory of Regulation [27], since the manipulation of the result is linked to the size of the organization, it is confirmed that the largest NSFs, being more visible, are more sensitive when responding to the regulations of the environment. Finally, although progress has been made in disclosure and in improvements in the financial situation of the Spanish NSFs, the present study shows that there is still a long way to go in terms of accountability.

Future studies could be aimed at expanding the time series, comparison with the NSFs of other countries, or trying to explain manipulation via other possible causes such as performance, transparency, the structures of the federative governing bodies, professionalization [66], and the adoption of techniques from the business sector [67]. For example, we suggest analysis of whether NSFs with higher quality accounting information also show

greater efficiency in resource management. For this purpose, the efficiency of NSFs will be measured using non-parametric techniques such as Data Envelopment Analysis or Färe Primont (see Torres et al., [5]). The results of this study can shed some light on the role of accountability as a value generator for organizations.

**Author Contributions:** Conceptualisation, J.C.G.; methodology, M.J.A. and J.C.G.; software, J.C.G.; analysis of results, J.C.G., E.M. and M.J.A.; writing—original draft preparation, J.C.G. and E.M.; writing—review and editing, J.C.G. and E.M. All authors have read and agreed to the published version of the manuscript.

**Funding:** This study has been carried out with the financial support of the Regional Government of Aragón/FEDER through the project GESPUBLICA S56-20R and the University of Zaragoza through Research Projects JIUZ-2018-SOC-01 and UZ2019-SOC-05.

**Acknowledgments:** We would like to appreciate the thoughtful and constructive advice provided by the reviewers, and especially the support of Néstor Le Clech from the Department of Economics and Business, National University of Quilmes, Argentina.

**Conflicts of Interest:** The authors declare no conflict of interest.

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
