# Peer review of "Financial Sustainability and Earnings Management in the Spanish Sports Federations: A Multi-Theoretical Approach"

_sustainability, doi:10.3390/su13042099_

Round 1
Reviewer 1 Report
see my attached report

Author Response
Thank you very much for your comments. We appreciate your thoughtful and constructive advice. Below, we try to respond to each of the issues raised in your review. In the new version of the paper, we have incorporated changes to address some of these suggestions and we believe that, as a result, the paper has been significantly improved. New text is in red and the text has been deleted is marked with a crossed-out line.

Reviewer 2 Report
The introduction, methodology, results and discussion are logic, well written, prepared and explained.
Minor comment: The manuscript should respect the Journal template.
Author Response
Dear Reviewer,
Thank you very much for your receptivity to our article.
Best regards.
Reviewer 3 Report
The paper is based on financial sustainability and earning management. However, some significant corrections are needed.
- What does it mean for sustainability in this study? Sustainability automatically indicates three pillars, then why does this study highlight financial separately? Explain it.
- Make an author contribution table to show the novelty of this study (See for reference, and you can refer to it for your study " Management of animal fat-based biodiesel supply chain under the paradigm of sustainability", “Combined effects of carbon emission and production quality improvement for fixed lifetime products in a sustainable supply chain management”, and “Environmental effect for a complex green supply-chain management to control waste: A sustainable approach”).
- An introduction should be extended, and the literature review should be extended more with recent references in this research direction.
- Bulky references should be deleted.
- The validity of data should be provided.
- The source of numerical data should be mentioned.
- Managerial insight should be mentioned.
- The proper extension of this paper with proper references should be added at the end of the conclusions.
Author Response

(The authors gave the same response as above.)

Round 2
Reviewer 3 Report
The paper can be accepted for publication.